

# Full genome sequence analysis of a 1-7-4-like PRRSV strain in Fujian Province, China

Jiankui Liu[1,2], Chunhua Wei[1,2], Zhifeng Lin[3], Wei Xia[1,2], Ying Ma[1,2], Ailing Dai[1,2] and Xiaoyan Yang[1,2]

[1] College of Life Sciences, Longyan University, Longyan, China
[2] Fujian Provincial Key Laboratory for the Prevention and Control of Animal Infectious Diseases and Biotechnology, Longyan University, Longyan, China
[3] College of Animal Science, Fujian Agriculture and Forestry University, Fuzhou, China

## ABSTRACT

PRRS virus (PRRSV) has undergone rapid evolution and resulted in immense economic losses worldwide. In the present study, a PRRSV strain named FJ0908 causing high abortion rate (25%) and mortality (40%) was detected in a swine herd in China. To determine if a new PRRSV genotype had emerged, we characterized the genetic characteristics of FJ0908. Phylogenetic analysis indicated that FJ0908 was related to 1-7-4-like strains circulating in the United States since 2014. Furthermore, the ORF5 sequence restriction fragment length polymorphism (RFLP) pattern of FJ0908 was 1-7-4. Additionally, FJ0908 had a 100 aa deletion (aa329–428) within nsp2, as compared to VR-2332, and the deletion pattern was consistent with most of 1-7-4 PRRSVs. Collectively, the data of this study contribute to the understanding of 1-7-4-like PRRSV molecular epidemiology in China.

Corresponding authors
Jiankui Liu, liujiankui@lyun.edu.cn
Chunhua Wei,
weichunhua02@163.com

## INTRODUCTION

Porcine reproductive and respiratory syndrome (PRRS) is a global viral swine disease and causing severe economic loss in the global pig industry (*Neumann et al., 2005*; *Zhou & Yang, 2010*; *Holtkamp et al., 2013*; *Gao et al., 2017*). PRRS virus (PRRSV), the causative agent of PRRS, is a small enveloped virus with positive-sense single-stranded RNA virus belonging to the Arteriviridae family in the Nidovirales order (*Benfield et al., 1992*; *Meulenberg, 2000*).

The PRRSV genome is about 15 kb in length and contains ten open reading frames (ORFs), ORF1a, 1b, 2a, 2b, 3, 4, 5a, 5, 6 and 7. ORF1a and ORF1b encode at least 16 non-structural proteins (nsps) (Nsp1α, Nsp1β, Nsp2-6, Nsp2TF, Nsp2N, Nsp7a, Nsp7b and Nsp8-12), while ORF2-ORF7 encode viral structural proteins: GP2a, E (2b), GP3, GP4, GP5a, GP5, M, and N (*Meulenberg, 2000*; *Wu et al., 2005*; *Firth et al., 2011*; *Fang & Snijder, 2010*; *Fang et al., 2012*).

PRRSV is characterized by extensive genetic variation. Based on global PRRSV classification systems, type 2 PRRSV strains in China can be classified into lineage 1

(NADC30-like), lineage 3 (QYYZ-like), lineage 5.1 (VR2332-like), and lineage 8.7 (JXA1-like) (*Gao et al., 2017*; *Liu et al., 2017b*; *Guo et al., 2018*). In Fujian Province, one of the largest livestock trading areas in China, herd movements across provinces and national borders are also common. Hence, multiple PRRSV types co-exist in swine herds. Recently, PRRSV isolates of the ORF5 restriction fragment length polymorphism (RFLP) 1-7-4 viruses emerged in America, causing dramatic abortion "storms" in the sow herd (*Alkhamis et al., 2016*; *Van Geelen et al., 2018*). Here, we report a genetic and phylogenetic analysis of PRRSV isolate FJ0908 belong to the ORF5 1-7-4 viruses in Fujian Province, China.

## MATERIALS AND METHODS

### Clinical case
In September 2018, severe reproductive and respiratory disease was observed in piglets in a farm. The affected pigs had respiratory distress, and high abortion rate (25%) and mortality (40%).

### Strain identification and nucleotide sequencing
PRRSV infection was confirmed by Real-time RT-PCR testing of the serum of affected pigs according to the manufacturer's instructions. The ORF5 sequence RFLP pattern was inferred according to Wesley et al. The complete genomic sequences of FJ0908 were amplified as described previously (*Zhang et al., 2018*). The PCR products were purified and cloned into pGEM-T Easy according to the manufacturer's instructions (Promega, Madison, WI, USA) and three recombinant clones of every fragment were sequenced by Ruibo Life Technologies Corporation (Beijing, China).

### Complete genomic sequence analysis
Fifty-four representative type 2PRRSV sequences in GenBank were utilized in phylogenetic analyses (Table 1). Multiplex sequence alignments were performed using CLUSTAL X (version 1.83) and the phylogenetic relationships were assessed by MEGA 6.0 as described (*Liu et al., 2017b*). The ORF5 sequences were classified according to the global PRRSV classification systems (*Shi et al., 2010*).

Recombination events were detected using Simplot v 3.5.1 and the boot scanning analysis was performed with a 200-bp window, sliding along the genome alignments with a step size of 20 bp.

## RESULTS

### Complete genomic sequence analysis
The genomes of FJ0908 (GenBank accession no. MK202794) was 15,112 nt in length, excluding the poly (A) tail at the 3′ end, 300 nt shorter than the genome of the prototypic VR2332. Genome alignments revealed that FJ0908 shared 83.6% identity with JXA1, 84.7% with VR2332, 82.2% with QYYZ, 86.3% with NADC30, 97.3–97.6% with 1-7-4 PRRSV family (IA/2014/NADC34, IA/2015/NADC35) and 98.7% with LNWK130 (Table 2).

The results also showed that 5′-UTR, ORF1a, ORF1b, ORFs 2-7 and the 3′-UTR of FJ0908 shared 97.2–98.2% nucleotide homology with 1-7-4 PRRSV family (IA/2014/NADC34,

**Table 1  PRRSV strains used in this study.**

| No. | Name | GenBank accession no. | Origin | No. | Name | GenBank accession no. | Origin |
|---|---|---|---|---|---|---|---|
| 1 | FJ0908 | MK202794 | China | 29 | SDSU73 | JN654458 | USA |
| 2 | FJZ03 | KP860909 | China | 30 | NADC31 | JN660150 | USA |
| 3 | FJY04 | KP860910 | China | 31 | NB/04 | FJ536165 | China |
| 4 | CH-1a | AY032626 | China | 32 | BJ-4 | AF331831 | China |
| 5 | JA142 | AY424271 | USA | 33 | MN184A | DQ176019 | USA |
| 6 | VR-2332 | U87392 | USA | 34 | MN184B | DQ176020 | USA |
| 7 | JXA1-R | R FJ548855 | China | 35 | HENNAN-XINX | KF611905 | China |
| 8 | JXA1 | EF112445 | China | 36 | NADC30 | JN654459 | USA |
| 9 | FJFS | KP998476 | China | 37 | JL580 | KR706343 | China |
| 10 | PA8 | AF176348 | Canadia | 38 | CHsx1401 | KP861625 | China |
| 11 | Em2007 | EU262603 | China | 39 | ISU30 | KT257977 | USA |
| 12 | GM2 | JN662424 | China | 40 | MN184C | EF488739 | USA |
| 13 | QYYZ | JQ308798 | China | 41 | NC/2014/ISU-3 | MF326990 | USA |
| 14 | HB-1(sh)/2002 | AY150312 | China | 42 | LV | M96262 | Netherlands |
| 15 | HB-2(sh)/2002 | AY262352 | China | 43 | NC/2014/ISU-4 | MF326991 | USA |
| 16 | HUN4 | EF635006 | China | 44 | IN/2014/ISU-5 | MF326992 | USA |
| 17 | OH/2014/ISU-6 | MF326993 | USA | 45 | Ingelvac ATP | DQ988080 | USA |
| 18 | NC/2015/ISU-11 | MF326998 | USA | 46 | RespPRRS MLV | AF066183 | USA |
| 19. | IA/2015/ISU-13 | MF327000 | USA | 47 | NC/2015/ISU-12 | MF326999 | USA |
| 20 | LNWK130 | MG913987 | China | 48 | IA/2015/NADC35 | MF326986 | USA |
| 21 | NCV-13 | KX192112 | USA | 49 | IA/2015/NADC36 | MF326987 | USA |
| 22 | NCV-23 | KX192116 | USA | 50 | IA/2015/ISU-10 | MF326997 | USA |
| 23 | NCV-25 | KX192118 | USA | 51 | IA/2014/NADC34 | MF326985 | USA |
| 24 | OH/2014/ISU-7 | MF326994 | USA | 52 | LNWK96-CN | MG860516 | China |
| 25 | IA/2014/ISU-8 | MF326995 | USA | 53 | NCV-21 | KX192115 | USA |
| 26 | IA/2015/NADC36 | MF326987 | USA | 54 | IA/2015/ISU-9 | MF326996 | USA |
| 27 | FJSD | KP998474 | China | 55 | HUB1 | EF075945 | China |
| 28 | IA/2014/ISU-2 | MF326989 | USA | 56 | IA/2015/ISU-14 | MF327001 | USA |

IA/2015/NADC35), which was higher than the homology shared with other representative strains, indicating that FJ0908 strain belonged to 1-7-4 PRRSV. ORF1a and ORF1b encode 16 nsps of PRRSV, Nsp1β and Nsp2 are the most variable protein products among these nsps (Table 2). ORFs 2 to 7 encode the PRRSV structural proteins, among these structural proteins, GP2, GP3, GP4, GP5a and GP5 exhibited the most variance (Table 2).

## Amino acid analysis of Nsp2

Nsp2 contains different deletions and insertions, as compared to VR2332 and is the most variable protein in PRRSV genome *Liu et al., 2017b*; *Li et al., 2011*). Strikingly, the nsp2 gene of the FJ0908 was 2,640 nt in length and encoded 880 aa, with a 100 aa deletion (aa329–428) within nsp2, as compared to VR2332, and the deletion pattern was consistent with most of 1-7-4 PRRSVs.

Liu et al. (2019), PeerJ, DOI 10.7717/peerj.7859

**Table 2  Detailed comparison of the full-length genomes of FJ0908 to other PRRSV reference strains.**

| | VR2332 | BJ-4 | JXA1 | HuN4 | FJFS | QYYZ | NADC30 | ISU30 | LNWK96 | IA/2014/NADC34 | IA/2015/NADC35 | LNWK130 |
|---|---|---|---|---|---|---|---|---|---|---|---|---|
| | Sublineage 5.1 | | Sublineage 8.7 | | Lineage 3 | | Sublineage 1.8 | | | | Sublineage 1.5 | |
| | | | | | Pairwise % Identity to FJ0908 (nt/aa) | | | | | | | |
| Nucleotides | | | | | | | | | | | | |
| Complete genome | 84.7 | 84.6 | 83.6 | 83.6 | 82.2 | 82.2 | 86.3 | 89.4 | 96.2 | 97.6 | 97.5 | 98.7 |
| 5′UTR | 93.6 | 93.6 | 92.5 | 92.5 | 91.0 | 91.5 | 95.7 | 92.9 | 96.3 | 97.9 | 97.3 | 97.9 |
| ORF1a | 82.0 | 81.9 | 80.6 | 80.7 | 78.6 | 78.6 | 82.8 | 87.8 | 96.3 | 97.2 | 97.2 | 98.0 |
| ORF1b | 86.6 | 86.6 | 86.2 | 86.2 | 85.8 | 85.6 | 88.8 | 86.2 | 98.2 | 98.2 | 98.1 | 99.2 |
| ORF2-7 | 87.2 | 87.1 | 85.9 | 86.0 | 84.9 | 84.9 | 89.4 | 88.3 | 93.1 | 97.6 | 97.2 | 99.6 |
| 3′UTR | 92.7 | 92.7 | 90.0 | 90.1 | 86.1 | 88.7 | 95.4 | 91.5 | 95.9 | 98.0 | 98.0 | 98.0 |
| nt 1–760 | 90.0 | 89.9 | 88.1 | 88.4 | 86.7 | 89.1 | 92.6 | 94.8 | 95.4 | 98.8 | 98.6 | 97.1 |
| nt 760–1,300 | 80.4 | 80.4 | 80.6 | 80.6 | 78.3 | 77.2 | 82.0 | 94.3 | 93.9 | 92.8 | 9.26 | 94.4 |
| Amino acids | | | | | | | | | | | | |
| NSP1α | 94.6 | 95.2 | 97.0 | 97.0 | 93.4 | 96.4 | 95.8 | 96.4 | 97.0 | 98.8 | 98.8 | 98.2 |
| NSP1β | 78.8 | 78.8 | 77.4 | 77.9 | 78.3 | 77.0 | 79.7 | 91.7 | 94.0 | 91.7 | 91.7 | 94.5 |
| NSP2 | 67.0 | 67.0 | 66.5 | 66.7 | 63.3 | 64.6 | 71.4 | 72.1 | 92.7 | 94.5 | 94.7 | 96.4 |
| NSP3 | 91.3 | 91.3 | 89.9 | 90.1 | 87.7 | 87.7 | 91.7 | 91.9 | 97.5 | 97.8 | 97.5 | 98.7 |
| NSP4 | 93.1 | 93.1 | 95.1 | 95.1 | 92.6 | 93.1 | 92.6 | 96.6 | 98.5 | 99.5 | 99.5 | 99.0 |
| NSP5 | 88.8 | 88.8 | 89.4 | 89.4 | 90.0 | 89.4 | 87.6 | 94.7 | 97.6 | 98.8 | 98.8 | 98.8 |
| NSP6 | 87.5 | 87.5 | 93.8 | 93.8 | 93.8 | 93.8 | 87.5 | 87.5 | 93.8 | 93.8 | 93.8 | 93.8 |
| NSP7 | 90.3 | 89.2 | 87.3 | 87.3 | 86.5 | 84.6 | 89.2 | 95.4 | 98.8 | 98.8 | 98.8 | 99.2 |
| NSP8 | 93.3 | 93.3 | 93.3 | 93.3 | 88.9 | 93.3 | 95.6 | 100 | 100 | 100 | 100 | 97.8 |
| NSP9 | 95.8 | 95.5 | 96.1 | 95.8 | 93.9 | 94.8 | 95.8 | 97.2 | 98.3 | 98.9 | 98.9 | 99.1 |
| NSP10 | 95.0 | 95.0 | 94.6 | 95.2 | 92.3 | 94.1 | 98.4 | 98.2 | 99.8 | 99.8 | 99.5 | 99.5 |
| NSP11 | 95.1 | 95.5 | 96.0 | 96.0 | 94.6 | 94.6 | 95.1 | 94.6 | 100 | 99.6 | 99.6 | 100 |
| NSP12 | 90.3 | 90.2 | 90.9 | 90.9 | 92.2 | 91.5 | 89.0 | 98.0 | 99.3 | 98.7 | 98.7 | 100 |
| ORF2a/GP2 | 85.5 | 86.3 | 85.9 | 85.5 | 80.1 | 82.8 | 83.6 | 84.0 | 97.7 | 98.4 | 96.9 | 99.6 |
| ORF2b/E | 87.8 | 89.0 | 86.3 | 86.3 | 80.8 | 91.8 | 90.4 | 90.4 | 97.3 | 100 | 98.6 | 98.6 |
| ORF3/GP3 | 80.7 | 80.7 | 79.9 | 79.5 | 80.7 | 79.9 | 81.5 | 81.9 | 89.4 | 94.5 | 93.3 | 99.2 |
| ORF4/GP4 | 86.0 | 84.8 | 86.5 | 88.8 | 88.8 | 86.5 | 94.4 | 92.1 | 91.6 | 94.4 | 94.4 | 99.4 |
| ORF5/GP5 | 87.0 | 87.0 | 86.0 | 86.5 | 85.5 | 85.5 | 90.5 | 88.0 | 90.0 | 97.5 | 97.5 | 98.5 |
| ORF5a | 91.3 | 91.3 | 84.8 | 84.8 | 84.8 | 87.0 | 95.7 | 93.5 | 95.7 | 100 | 100 | 97.8 |
| ORF6/M | 95.4 | 94.8 | 93.7 | 93.7 | 95.4 | 94.8 | 94.8 | 93.7 | 95.4 | 97.7 | 97.7 | 100 |
| ORF7/N | 91.1 | 90.2 | 90.2 | 90.2 | 87.8 | 87.8 | 95.9 | 93.5 | 9.19 | 96.7 | 95.1 | 99.2 |

## Antigenic analysis of GP2-GP5

The antigenic regions (ARs) and glycosylation sites within the GP2, GP3, GP4, and GP5 proteins of FJ0908 were predicted and compared to OH/2014/ISU-7 IA/2014/ISU-8, IA/2014/NADC34, IA/2015/NADC35, IA/2015/NADC36, LNWK96, LNWK130, NV-21, NV-25, VR2332, CH-1a, JXA1 and NADC30.

In GP2, two antigenic regions (AR 41–55 and AR123–135) were confirmed in type 2 PRRSV (*De Lima et al., 2006*). The predicted AR at aa 41–55 were highly conserved and no aa substitution was detected in AR123–135. In GP3, four predicted antigenic regions (AR32–46, AR 51–105, AR 111–125, and AR 137–159) were proven (*De Lima et al., 2006*; *Zhou et al., 2006*; *Wang et al., 2014*). The AR comprising aa32–46, aa51–105, aa111–125, and aa137–159 of FJ0908 was most similar to 1-7-4 representative PRRSVs including LNWK130, but differed from VR2332, CH-1a, JXA1 and NADC30. GP4 has one predicted AR at aa51–65 (*De Lima et al., 2006*). FJ0908 had 1-2 aa substitutions as compared to the 1-7-4 representative PRRSVs, but had 2-6 aa substitutions as compared to VR2332, CH-1a, JXA1 and NADC30 and LNWK96. Additionally, no aa substitution was detected in AR51–65 between FJ0908 and LNWK130. The putative glycosylation sites in GP2-4 were completely conserved among the investigated strains, except for GP2 of isolate IA/2015/NADC36 that had one substitution $N^{184}D$.

GP5 is the major envelope protein encoded by ORF5 and is the most variable PRRSV protein. Sequences alignments of GP5 revealed that FJ0908 shares 87.0%, 86.0–86.5%, 85.5%, 88.0–90.5% and 97.5% amino acid identity with VR-2332-like (VR2332 and BJ-4), JXA1-like (JXA1 and HuN4), QYYZ-like (FJFS and QYYZ), NADC30-like strains (NADC30 and ISU30), and 1-7-4-like (IA/2014/NADC34, IA/2015/NADC35), respectively (Table 2). Furthermore, the restriction sites analysis showed that the ORF5 RFLP of FJ0908 has the same 1-7-4 pattern [1 (MluI = 0 sites), 7 (HindII = nt 88, 219, 360), 4 (SacII = nt 24, 555)] (Fig. S1).

GP5 has six ARs (AR1–15, AR27–35, AR37–51, AR149–156, AR166–181, and AR192–200) (*De Lima et al., 2006*; *Zhou et al., 2009*). In GP5, the N-terminus ARs (AR1-15 and AR27-35) were very variable among all strains and only three aa substitutions ($K^4N$, $Q^{13}R$ and $L^{15}P$) were found in the two antigenic regions between FJ0908 and LNWK130, whereas the other four ARs (AR 37–51, AR 149–156 , AR 166–181, and AR 192–200) were conserved. GP5 had differential predicted N-glycosylation among PRRSV strains. FJ0908 possessed five predicted sites (N32, N44, N51, N57 and N59). Although N-glycosylation at N57 was not a novel finding, N57 was detected in most of the 1-7-4 sequences. The results also revealed that FJ0908 had the same N-glycosylation pattern as LNWK96 and LNWK130.

## Phylogenetic analysis

PRRSV ORF5 is the most variable and has been used as a marker of PRRSV genetic variability. Based on global PRRSV classification systems, type 2 PRRSV was divided into nine monophyletic lineages (1-9) and lineage1 was further classified into nine sublineages (1.1–1.9) (*Shi et al., 2010*; *Guo et al., 2018*).  The ORF5-based phylogenetic tree showed that FJ0908, as well as 1-7-4 isolates including LNWK130, were clustered in sublineage 1.5

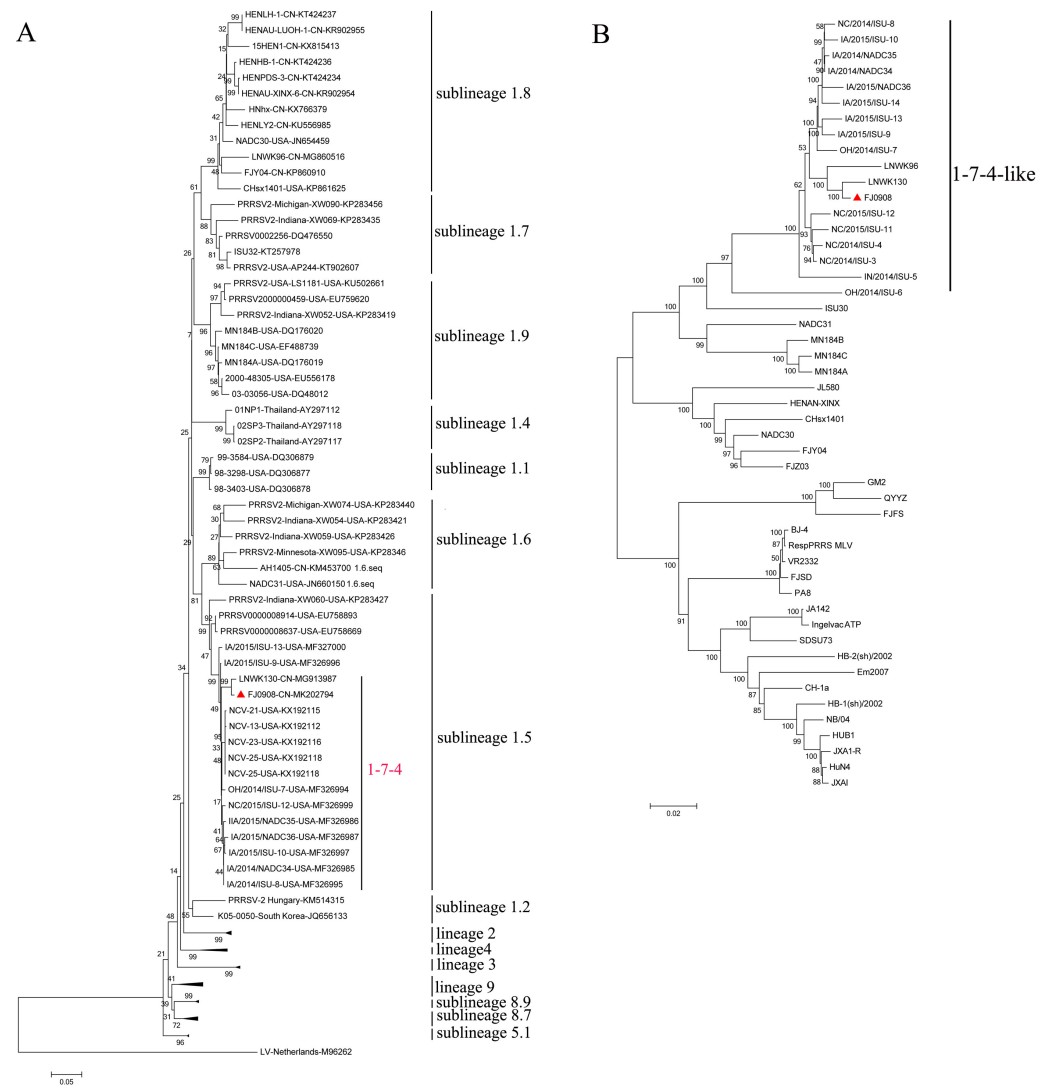

**Figure 1  Phylogenetic tree based on the ORF5 genes (A) and full length (B) of the FJ0908 and reference viruses.** Reliability of the tree was assessed by bootstrap analysis of 1,000 replications. Our representative isolate FJ0908 were marked with the red triangle (▲). Lineage 1 PRRSVs are divided into nine sublineages.

(Fig. 1A). Whole genome phylogenetic analysis also indicated that FJ0908 was most closely related to a genetic cluster in 1-7-4-like lineage 1 (Fig. 1B).

## Recombination analysis

To test for possible recombinant events within FJ0908 strain, we performed recombinant detection using SimPlot v3.5.1 software. From the similarity plot, two recombination breakpoints within the FJ0908 genome were identified, which were located in Nsp1 (nt 760 and nt 1,300) (Fig. 2A). To further confirm the putative recombination events, phylogenetic trees for each of the sequence regions identified during the analysis were generated, we identified two recombination breakpoints located in nsp1 (nt 760 and nt 1,300)

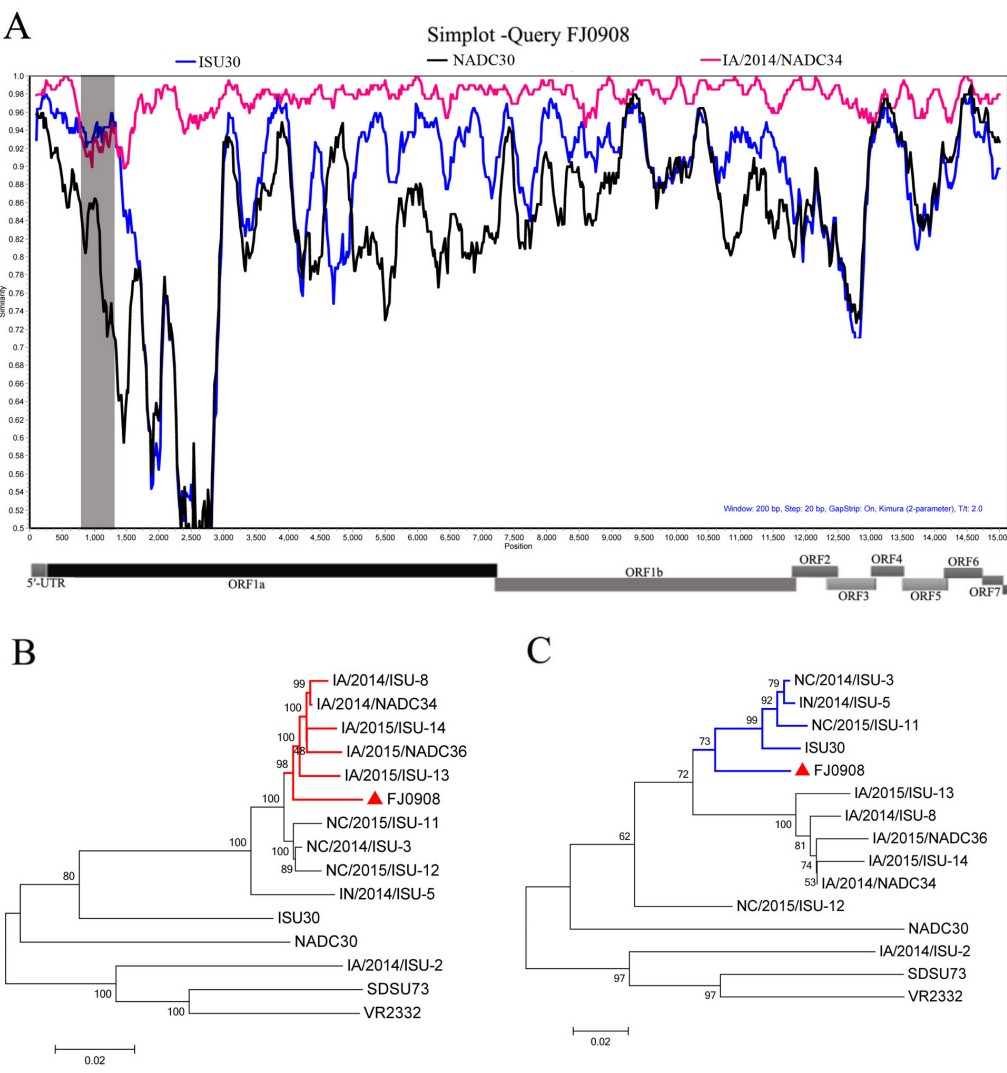

**Figure 2 Characterization of the supported recombinant events between FJL0908 and representative PRRSV lineages.** (A) Similarity plot and bootscan analyses of FJ0908 by SimPlot. The *y*-axis indicates the percentage similarity between the parental sequences and the query sequence. Phylogenetic trees based on major parental regions (nt 1-760 and nt 1301-15534) (B) and the minor parental region (nt 760–1300) (C). The major parental group (1-7-4 viruses, reference strain IA/2014/NADC34) is shown in red, while the minor parental groups (reference strain ISU30) are shown in blue, respectively.

(Figs. 2B and 2C). The two breakpoints separated the genome of FJ0908 into 3 regions. For FJ0908, the region between the breakpoints (nt 760–1,300) is closely related to ISU30 strain, the two regions between the breakpoints (nt 1–759 and nt 1,301–15,534) are closely related to IA/2014/NADC34. Collectively, the above results suggested that FJ0908 derived from recombination between IA/2014/NADC34 and ISU30 (Fig. 2). Moreover, LNWK130 strain (1-7-4-like PRRSV) firstly identified in Liaoning Province, China was also deriving from the recombination of 1-7-4 isolates and ISU30 (*Zhang et al., 2018*). However, the recombination pattern was different between LNWK130 and FJ0908. For LNWK130,

the two breakpoints (nt 760–1,300 and nt 1–759) are closely related to ISU30 strain, one breakpoints (nt 1301–15534) is closely related to IA/2014/NADC34. Additionally, sequences alignments of the recombination region (nt 1-759) revealed that FJ0908 shares 98.8%, 94.8% and 92.6% nucleotide identity with IA/2014/NADC34, ISU30 and NADC30, respectively, in contrast, LNWK130 shares 95.8%, 95.9% and 93.1% nucleotide identity with IA/2014/NADC34, ISU30, and NADC30, respectively. Sequences alignments of the recombination region (nt 760-1,300) revealed that FJ0908 shares 92.8%, 94.3% and 82.0% nucleotide identity with IA/2014/NADC34, ISU30 and NADC30, respectively, similarly, LNWK130 shares 89.4%, 93.0% and 80.2% nucleotide identity with IA/2014/NADC34, ISU30, and NADC30, respectively. For all of the nsp sequences and structural proteins, the most variable regions were found in nsp1β, nsp3, nsp5, nsp6, nsp8, GP3 and GP5 between FJ0908 and LNWK130.

## DISCUSSION

PRRSV causes major economic losses in swine industry since 1990s. Notably, PRRSV continues to expand its genetic diversity. According to *Shi et al. (2010)*, type 2 PRRSV was classified into nine monophyletic lineages based on ORF5 and extensive genetic variation exists among strains within each lineage. It is hard to define the PRRSV homologous, heterologous virus and pathogenic biotype only focused on single gene analysis. To classify and infer the likely pathogenic biotype, RFLP patterns of ORF5 for type 2 strains is standard approach for veterinarians (*Wesley et al., 1998*). The RFLP pattern 1-7-4 emerged in the US and has become prevalent since 2014, this nomenclature has been associated with severe disease in herds leading to significant economic losses (*Van Geelen et al., 2018*). In the present study, FJ0908 was isolated in a farm with high abortion rate and mortality in sows, the restriction sites analysis revealed that the ORF5 RFLP of FJ0908 has the 1-7-4 pattern.

Comparison to PRRS sequences in GenBank indicated FJ0908 belonged to 1-7-4-like PRRSV. The genomic regions with the highest variation were found in Nsp1β, Nsp2, ORF2, ORF3, ORF4, ORF5a and ORF5, the lowest variation were found in Nsp1α, Nsp8–12, and ORF6 (Table 2). FJ0908 had 100 aa deletions within Nsp2 (corresponding to position 328–427 of VR2332 nsp2), as compared to the reference strain VR2332, and the deletion pattern was consistent with 1-7-4 viruses.

The pathogenesis of PRRSV has been linked to the N-glycosylation motifs at certain sites of GP2-GP5 by acting as a glycan shield against minimizing the viral neutralizing antibody response (*Wissink et al., 2004*; *Ansari et al., 2006*; *Faaberg et al., 2006*; *Das et al., 2011*; *Delisle et al., 2012*; *Wei et al., 2012*). Many reports have also suggested that N-glycosylation motifs in GP5 of PRRSV is important for viral infectivity and viral immune evasion (*Wissink et al., 2004*; *Ansari et al., 2006*; *Jiang et al., 2007*; *Delisle et al., 2012*; *Wei et al., 2012*). In the present study, GP5 contained five predicted N-glycan motifs in FJ0908: N32, N44, N51, N57 and N59. More interestingly, the Chinese strains LNWK96, LNWK130 and FJ0908 have an additional N-glycan at 59 compared to 1-7-4 isolates in the United States.

Recombination may play an important mechanism in generating genetic diversity in PRRSV (*Liu et al., 2011*; *Murtaugh et al., 2010*). Most of the 1-7-4 PRRSV isolates may

be most potentially derived from different recombination patterns occurring among the local strains in the United States (*Van Geelen et al., 2018*). Recently, 1-7-4-like PRRSV strains, LNWK130 isolated in Liaoning Province, China was reported to originate from recombination events between IA/2014/NADC34 and ISU30. Currently, recombination events involving NADC30-like PRRSV strains and other PRRSV strains frequently occurred in China (*Zhao et al., 2015*; *Zhang et al., 2016*; *Bian et al., 2017*; *Liu et al., 2017a*; *Liu et al., 2017c*; *Zhao et al., 2017*; *Wang et al., 2018*; *Zhou et al., 2018*; *Liu et al., 2019*). To test for possible recombinant events within FJ0908 strain, we performed recombinant detection using SimPlot v3.5.1 software. Recombination analysis performed with the available full-length genome sequences revealed FJ0908 maybe originate from recombination events between IA/2014/NADC34 and ISU30. Although FJ0908 and LNWK130 maybe drive from recombination events between IA/2014/NADC34 and ISU30, the recombination pattern of two strains were different. Two recombination breakpoints were identified in nsp1 (nt 760 and nt 1,300) in FJ0908 strain, whereas one recombination breakpoint in nsp2 (nt 1480) in LNWK130, suggesting the ancestor of FJ0908 and LNWK130 were most probably transported from different region of United States.

In conclusion, 1-7-4-like PRRSV was also detected in Fujian Province of China besides Liaoning Province. Therefore, effective strategy should be taken to control 1-7-4-like PRRSV and to monitor herd movements.

## CONCLUSION

In summary, we thoroughly analyzed a new sublineage of PRRSV strain FJ0908 isolated from Fujian Province, China on the basis of a comprehensive study with the full-length genome. These novel sublineage 1.5 virus is closely related to the ORF5 RFLP 1-7-4 strains. Phylogenetic and molecular evolutionary analyses indicated that FJ0908 originated from a natural recombination event between IA/2014/NADC34 and ISU30. Our data enhance our understanding of the PRRSV evolution in China.

### Funding

This study was supported by the Leading Project Foundation of Science Department of Fujian Province (2018N0022), the Major Project of Science and Technology Program of Fujian Province, China (2019NZ09005) and the Fujian Natural Science Foundation, Fujian Province, China (2016J01168). The funders had no role in study design, data collection and analysis, decision to publish, or preparation of the manuscript.

### Grant Disclosures

The following grant information was disclosed by the authors:
Leading Project Foundation of Science Department of Fujian Province: 2018N0022.
Major Project of Science and Technology Program of Fujian Province, China: 2019NZ09005.
Fujian Natural Science Foundation, Fujian Province, China: 2016J01168.

## Competing Interests

The authors declare there are no competing interests.

## Author Contributions

- Jiankui Liu conceived and designed the experiments, analyzed the data, contributed reagents/materials/analysis tools, prepared figures and/or tables, authored or reviewed drafts of the paper, approved the final draft, secured funding.
- Chunhua Wei and Zhifeng Lin performed the experiments, contributed reagents/materials/analysis tools, prepared figures and/or tables, authored or reviewed drafts of the paper, approved the final draft.
- Wei Xia and Ying Ma performed the experiments, contributed reagents/materials/analysis tools, prepared figures and/or tables, approved the final draft.
- Ailing Dai and Xiaoyan Yang analyzed the data, contributed reagents/materials/analysis tools, approved the final draft.

## DNA Deposition

The following information was supplied regarding the deposition of DNA sequences:
   The sequence is available at GenBank: MK202794.

## Data Availability

   The sequence is available as a Supplemental File and at GenBank.

## Supplemental Information

Supplemental information for this article can be found online at http://dx.doi.org/10.7717/peerj.7859#supplemental-information.

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
