# Peer review of "Full genome sequence analysis of a 1-7-4-like PRRSV strain in Fujian Province, China"

_PeerJ, doi:10.7717/peerj.7859_

## Round 0.1 · original submission · Minor Revisions

In addition to their other comments and suggestions, both reviewers suggest that providing a more thorough analysis comparing the characteristics of FJ0908 to the isolate LNWK130 would be in order given their close phylogenetic relationship. Also, please be sure to properly italicize the virus taxon names.

Reviewer 1 ·

Basic reporting

No comment

Experimental design

no comment

Validity of the findings

no comment

Additional comments

Authors report the molecular characterization of an PRRSV isolate named FJ0908 in Fujian Province, China. I recommend major changes to accept the manuscript. Although the methodological approach is correct, the authors should indicate what is the difference between FJ0908 with the strain LNWK130 previously reported and characterized by Zhang et al., 2018. LNWK130 is an emergent strain from sublineage 1.5 and derived from the recombination of 1-7-4 isolates which were isolated in the United States. LNWK130 was identified in Liaoning Province and for the first time in China (Zhang et al., 2018: Emergence of novel porcine reproductive and respiratory syndrome viruses (ORF5 RFLP 1-7-4 viruses) in China. Vet Microbiol. 222:105-108).
If both strains (FJ0908 and LNWK130) have the same recombination pattern, this work, in the way it is proposed, does not have great originality. The important point would be that this emerging strain was found for the first time in Funjian province, clarifying the differences with the strain LNWK130 (if there are any).

Reviewer 2 ·

Basic reporting

1. Figure 2 is too small to see clearly. Need to redesign/reorganize this figure so it's readable when printed on a paper.
2. In Table 1, LNWK96 was isolated in China, not USA. This should be corrected. Other isolates should be double checked as the reviewer cannot check for you one by one.
3. Line 59, "more than ten open read frames", but line 60 only listed 10 ORFs. Either remove the "more than" on line 59 or list more than 10 ORFs.
4. Line 71, 1-7-4 should not be called a lineage here, especially lineages, 1, 3, 4.1, 8.7 were just introduced in the same paragraph. Calling 1-7-4 in this context may mislead the readers to think 1-7-4 is a true lineage in parallel to them, while actually it belongs to lineage 1 sublineage 1.5.
5. Remove the period on line 84.
6. Lines 123-124, "four proven antigenic regions were proven"? should the first proven be changed to "predicted"?
7. Line 169, add "is" between "It" and "hard".

Experimental design

1. The previous study by Zhang et al. and the isolate LNWK130 should be compared and discussed more in the current study. Due credit should be given to the previous study, as many of the analyses (including the PCR method) are either very similar to the Zhang et al. study or used the method in Zhang et al.

Validity of the findings

no comment.

---

## Round 0.2 · Major Revisions

As Reviewer 1 suggests, this report needs to distinguish the sequence and properties of the FJ0908 isolate of PRRSV from the closely related isolate, LNWK130 in order to justify a publication describing the unique features of FJ0908. The review suggests a number of ways that the manuscript can be improved by improving the comparative analysis. Also, please be sure to check the labels of the figures for errors. For example, in figure 4C, two branches are labeled IA/2015/NADC36 while the isolate IA/2014/NADC34 does not appear.

Reviewer 1 ·

Basic reporting

no comment

Experimental design

see general comments

Validity of the findings

see general comments

Additional comments

The corrections made in the manuscript are not enough to show the differences between LNWK130 and FJ0908. I think that the central point of this article is to focus on the comparison of these two strains since LNWK130 was identified for the first time in Liaoning Province China and shared a high sequence identity with FJ0908. Authors sould highlight that this emergent strain now also circulates in another province of China, Fujian Province.
To improve this point, authors should:
-Deepen the information provided in table 2 regarding the nucleotide changes between FJ0908 and LNW130.For example, focus on changes in the recombination zone.
-Replace the analysis of figure 2 by amino acid comparison between FJ0908 and LNW130.
-Perform the recombination study for LNWK130 with the same methodology and the same sequences used for the study of FJ0908 and compare the recombination breakpoints. This can be shown in a complementary figure.
Minor comments:
Eliminate figure 1, it is not necessary since the same information is published in Zhang et al. 2018
Show the recombination zone in gray in Figure 4 as it was in the previous version.
Problem with spaces between words in the new PDF version of the manuscript

---

## Round 0.3 · Major Revisions

Before I can send out your revised manuscript for a second review, please address the critique I previously provided as Editor for your manuscript. The following problem has not been resolved: In figure 2C (previously 4C), two branches are labeled IA/2015/NADC36 while the isolate IA/2014/NADC34 does not appear. In addition, please be sure to check all of the labels in all figures to be sure they are all correct. Thank-you.

---

## Round 0.4 · Minor Revisions

Please carefully read and respond to Reviewer 1's critique for revision 3 of your manuscript. In your response to the last round of critiques, you indicated that "Considering the Reviewer’s suggestion, we performed the recombination study for LNWK130 with the same methodology and the same sequences used for the study of FJ0908 and compare the recombination breakpoints in figure 2 and manuscript.". But the current version of figure 2 does not contain an analysis of LNWK130. This is at the heart of the current critique from Reviewer 1. While recombination for LNWK130 is discussed in the text, one possibility that would help clear up Reviewer' 1's critique would be to include a panel for LNWK130 in figure 2. Once this concern has been addressed, we will be happy to accept your manuscript for publication.

Reviewer 1 ·

Basic reporting

see general comments

Experimental design

see general comments

Validity of the findings

see general comments

Additional comments

The manuscript is ready to publish correcting some minor points: It is very important that LNWK130 appear in the analysis of figure 1 but it does not have to appear in the recombination study (Figure 2b and 2c). This brings confusion and suggests that it is the same type of recombinant. Others recombinant strains should not be included in a recombination analysis. However, the comparison is neccesary since LNWK130 is a previously identified recombinant strain in China that has a high similarity with the strain identified in this work. In line 165 authors should emphasize that LNWK130 is a recombinant strain too but is something different to FJ0908 and then compare the breakpoints. Authors must indicate the reference or put "data not shown" if they performed the analysis.

I don't understand what the authors wanted to put in the conclusión: In conclusion, 1-7-4-like PRRSV was also detected in Fujian Province of China except Liaoning Province. Except?

---

## Round 0.5 · accepted · Accept

It was a pleasure to work with you during the review process of your manuscript.